# Stiffness-switchable DNA-based constitutional dynamic network hydrogels for self-healing and matrix-guided controlled chemical processes

Liang Yue[1], Shan Wang[1], Verena Wulf[1] & Itamar Willner [1]*

Constitutional dynamic networks (CDNs) attract interest as signal-triggered reconfigurable systems mimicking natural networks. The application of CDNs to control material properties is, however, a major challenge. Here we report on the design of a CDN consisting of four toehold-modified constituents, two of which act as bidentate units for chain-elongating, while the other two form a tetradentate structure acting as a crosslinking unit. Their hybridization yields a hydrogel of medium stiffness controlled by the balance between bidentate and tetradentate units. Stabilization of the tetradentate constituent by an auxiliary effector up-regulates the crosslinking unit, yielding a high-stiffness hydrogel. Conversely, stabilization of one of the bidentate constituents by an orthogonal effector enriches the chain-elongation units leading to a low-stiffness hydrogel. Using appropriate counter effectors, the hydrogels are reversibly switched across low-, medium- and high-stiffness states. The hydrogels are used to develop self-healing and controlled drug-release matrices and functional materials for operating biocatalytic cascades.

[1] Institute of Chemistry, The Hebrew University of Jerusalem, 91904 Jerusalem, Israel. *email: willnea@vms.huji.ac.il

The design of constitutional dynamic networks (CDNs) that mimic functions of natural networks attracts growing research efforts[1–4]. A simple [2 × 2] network, CDN I, includes four dynamically equilibrated constituents AA', BB', AB' and BA', which can be reversibly reconfigured in their compositions in the presence of appropriate auxiliary triggers. Subjecting this CDN to trigger $T_1$ that stabilizes AA' in the form of AA'-$T_1$ results in the reconfiguration of CDN I into a second CDN, CDN II, where AA' is upregulated, AB' and BA' are downregulated, and consequently BB' (not sharing common components with AA') is concomitantly upregulated. In addition, treatment of the resulting CDN II with the counter trigger $T_1'$, which removes the stabilizer from AA'-$T_1$, recovers the original CDN I. Similarly, subjecting CDN I to an orthogonal trigger $T_2$ that stabilizes AB' in the form of AB'-$T_2$, leads to the adaptive reconfiguration of the CDN to a third CDN, CDN III, where AB' is upregulated, accompanying with the downregulation of AA' and BB' and the concomitant upregulation of BA' that does not share common components with AB'. Substantial progress in the design of dynamic networks of low-molecular-weight constituents that reveal triggered, switchable, programmed, and adaptive equilibration properties, was reported[5], e.g., a dynamically equilibrated hydrazone/acylhydrazone system undergoing orthogonal reconfiguration in the presence of metal ions or light as triggers[6]. Different triggers, such as light[6–8], temperature[9], pH[9], electric fields[10], or solvents[11], were applied to reconfigure the CDNs. Nonetheless, these studies emphasized the difficulties in advancing the field of CDNs due to: (i) The lack of a versatile chemical platform to design the CDNs and the need to reengineer the molecular principles to assemble and trigger each of the CDNs are certain disadvantages. (ii) The difficulties in designing principles to intercommunicate between CDNs and to enhance the structural complexity of the constituents are drawbacks. (iii) The lack of versatile principles to combine the adaptive and programmable features of CDNs with emerging functions of the systems is a major challenge.

In a series of recent reports, we have introduced nucleic acids as versatile units to construct constitutional dynamic network systems of enhanced complexities[12–17]. In these systems, we made use of several fundamental features of nucleic acids that turned the approach into an integrated module to assemble CDNs: (i) The base sequence of the nucleic acids encodes substantial structural and functional information into the biopolymers, e.g., duplexes[18,19], triplexes[20,21], G-quadruplexes[22,23], and allows the programmed predesign of the constituents comprising the CDNs. (ii) The control over the stabilities of the nucleic acid constituents by means of auxiliary triggers, such as strand displacement[24,25], pH[26], metal ions[27,28], or light[29,30], provides a means to trigger the reconfiguration of the CDNs. (iii) Specific nucleic acid sequences reveal catalytic functions (DNAzymes)[31,32]. The conjugation of DNAzyme units to the constituents provides catalytic "tools" to quantify the contents of the constituents in the CDNs and to follow the kinetics of the dynamic reequilibration of the systems, and internally integrated catalysts that allow the intercommunication between CDNs. Indeed, we presented the adaptive, reversible reconfiguration of CDNs using a strand displacement mechanism in the presence of fuel/anti-fuel strands[12], the formation and dissociation of G-quadruplexes in the presence of K+ ions/crown ether[12,13], and the light-induced stabilization/destabilization of duplexes, e.g., *trans–cis* photoisomerization of azobenzene[14]. In addition, the adaptive reequilibration of intercommunicating CDNs[15], the triggered activation of feedback-driven CDNs[16], and the design of the CDNs of enhanced complexity ([3 × 2] or [3 × 3])[17] that reveal adaptive and hierarchical control over the compositions of the CDNs, were demonstrated. The real challenge in advancing nucleic acid-based CDNs involves, however, the use of these programmed assemblies for practical applications. Recently, CDN-guided aggregation of Au nanoparticles and the control over their catalytic functions, e.g., the oxidation of dopamine by $H_2O_2$, were reported[33]. In addition, previous studies have reported on the formation of gels through the dynamic covalent selection of C=C/C=N[34] or guanine quartets[35] as gelation subunits. Nonetheless, these systems underwent only sol-to-gel transitions and did not reveal any embedded functions.

Here, we report on the assembly of nucleic acid-based, CDN-guided hydrogel materials. We demonstrate that the adaptive, switchable and reversible transitions of the CDNs across three different states lead to hydrogels exhibiting three different stiffness properties: high stiffness, medium stiffness and low stiffness. These unique properties of the hydrogels are used to yield self-healing hydrogels, stimuli-controlled release of loads (drugs) from the hydrogels, and a switchable activation of enzyme cascades integrated into the hydrogels. That is, beyond the fundamental characterizations of the CDN-driven switchable stiffness properties of the hydrogels, we introduce the coupling between constitutional dynamic networks and material chemistry. It should be noted that in contrast to previously reported stimuli-responsive DNA-based hydrogels, the present system represents an approach that applies CDNs for the controlled interconversion of nucleic acid network hydrogels that yield three different states of stiffness. In addition, beyond the important material properties of the resulting stiffness-switched hydrogels, the study introduces a basic scientific advance by providing an optical (fluorescence) means to correlate between the molecular compositions of the hydrogels and their bulk mechanical properties.

## Results

**CDN-guided hydrogels.** Figure 1 describes the compositions of the DNA-based CDN systems and discusses the concept of reversibly controlling the stiffness of the hydrogels. CDN X includes four dynamically equilibrated constituents AA', BB', AB' and BA', where the constituent pairs AA'/BB' and AB'/BA' exhibit agonist relations (not sharing common components). AA' and BB' are Y-shaped supramolecular structures of three nucleic acid strands (A, A' and $L_1$ for AA'; B, B' and $L_2$ for BB'), which act as bidentate units offering two single-stranded toehold tethers (c and d in AA'; c' and d' in BB'). The other two constituents AB' and BA' reveal antagonistic relations to AA' and BB' (sharing common components with AA' and BB'). These two constituents form a supramolecular structure of enhanced complexity, composed of six interlinked nucleic acid strands (A, B, A', B', $L_1$, and $L_2$), which acts as a tetradentate unit (AB'/BA') offering four single-stranded toehold tethers (c, d, c', and d'). Note that $L_1$ and $L_2$ do not participate in the equilibrium of the network. The single-stranded toehold tethers associated with the four constituents exhibit complementary relations of cc' and dd'. In addition, each of the constituents includes a quasi-loop single-stranded domain. These domains allow the hybridization of the respective constituents with auxiliary triggers and thus the control over their stabilities. The combination of these four constituents leads to a crosslinked all-DNA CDN hydrogel X, where the tetradentate unit AB'/BA' acts as a hydrogel crosslinking unit, and the bidentate units AA' and BB' act as chain-elongation units. The balance between the bidentate and tetradentate units controls the degree of crosslinking and thus the stiffness of the resulting hydrogel. Note that the bidentate units (AA' and BB') act as terminating sites for the crosslinked polymerization, while the tetradentate unit (AB'/BA') increases the branching and crosslinking of the hydrogel. That is, the degree of crosslinking and thus the stiffness of CDN hydrogel X are controlled by the contents of the constituents in CDN X. Treatment of hydrogel X with

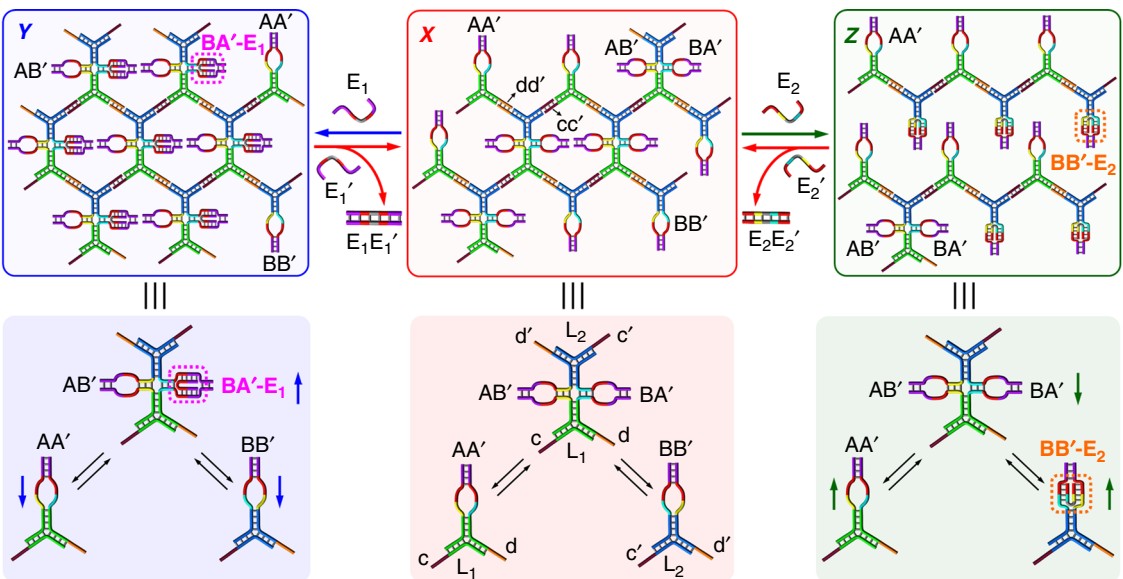

**Fig. 1** Schematic control over the compositions and stiffness of hydrogels guided by constitutional dynamic networks. The CDN hydrogel of medium stiffness, X, is formed by the crosslinking of the toehold-modified constituents AA', BB', AB', and BA'. The $E_1$-guided stabilization of BA' in CDN hydrogel X leads to the reequilibration of hydrogel X to hydrogel Y with an enhanced stiffness due to the triggered increase of the tetradentate crosslinking unit AB'/BA'. The $E_2$-guided stabilization of BB' in CDN hydrogel X leads to the transition of hydrogel X to Z exhibiting a lower stiffness due to the triggered depletion of the tetradentate crosslinking unit AB'/BA'. Subjecting the resulting hydrogel Y or Z to the counter trigger $E_1$' or $E_2$', respectively, regenerates the original hydrogel X

trigger $E_1$ stabilizes BA' via the hybridization of its single-stranded quasi-loop with $E_1$ in the form of BA'-$E_1$. This leads to an adaptive reequilibration of CDN X to CDN Y, where BA'-$E_1$ and its agonist AB' are upregulated and its antagonists AA' and BB' are downregulated. The resulting polymerization and cross-linking of the reconfigured constituents in CDN Y lead to hydrogel Y. In this hydrogel configuration, the content of the tetradentate crosslinking unit is increased, while the content of the bidentate units is decreased. That is, hydrogel Y reveals enhanced crosslinking and thus a higher stiffness as compared to hydrogel X. Treatment of hydrogel Y with the counter trigger $E_1$' leads to a strand displacement of $E_1$ by forming the duplex $E_1E_1$' and to the restoration of hydrogel X, exhibiting the original stiffness. In analogy, treatment of hydrogel X with an orthogonal trigger $E_2$ results in the hybridization of BB' with $E_2$. The stabilization of BB' in the form of BB'-$E_2$ leads to the reequilibration of CDN X to CDN Z, where BB'-$E_2$ and its agonist AA' are upregulated and its antagonists AB' and BA' are downregulated. This leads to hydrogel Z that includes a higher content of the bidentate units (AA' and BB') and a lower content of the tetradentate crosslinking unit AB'/BA', as compared to hydrogel X. That is, hydrogel Z is anticipated to reveal a lower stiffness as compared to hydrogel X. Subjecting hydrogel Z to the counter trigger $E_2$' leads to a strand displacement of $E_2$ by forming $E_2E_2$' and to the recovery of hydrogel X. That is, the interaction of CDN hydrogel X with trigger $E_1$ or $E_2$ allows to switch the stiffness of the hydrogel, and the application of the counter trigger $E_1$' or $E_2$' allows the recovery of the original-stiffness state of the hydrogel.

**Reversible control of stiffness properties of CDN hydrogels.** Figure 2a shows the stiffness properties of the three CDN hydrogels. The Young's modulus of hydrogel X corresponds to $13.18 \pm 0.29$ kPa, the Young's modulus of hydrogel Y is substantially higher, $23.45 \pm 0.43$ kPa, while hydrogel Z exhibits a significantly lower Young's modulus of $3.75 \pm 0.03$ kPa (Supplementary Table 1, standard errors were derived from 100

indentations). These Young's moduli are consistent with the degree of crosslinking of the hydrogels controlled by the compositions of the CDNs (*cf.* Fig. 1). Furthermore, the stiffness changes of the hydrogels are reversible. Supplementary Fig. 1 and Table 1 show the reversible stiffness changes upon switching hydrogel X to hydrogel Y or Z using trigger $E_1$ or $E_2$ and back in the presence of the counter trigger $E_1$' or $E_2$', respectively. In addition, we found that the $E_1$-triggered or $E_2$-triggered stiffness properties of hydrogel X can be programmed by the concentrations of the effectors, Fig. 2b. With increasing the concentration of effector $E_1$, the stiffness of the resulting hydrogel increases. Accordingly, with increasing the concentration of $E_2$, the stiffness of the resulting hydrogel decreases. While the microindentation results represent switchable surface properties, it is interesting to address the bulk mechanical properties (storage and loss moduli, G' and G") of the different hydrogels, and specifically to follow the time-dependent mechanical property changes of hydrogel X subjected to effector $E_1$ or $E_2$ to yield hydrogel Y or Z, respectively. The G' and G" values of hydrogel X correspond to G' = 145.6 Pa and G" = 11.9 Pa. The $E_1$-triggered transition of hydrogel X to Y yields higher storage and loss moduli, G' = 264.9 Pa and G" = 47.5 Pa. The treatment of hydrogel X with effector $E_2$ yields the soft hydrogel Z with G' = 53.7 Pa and G" = 6.5 Pa (Supplementary Fig. 2). CDN hydrogel Z, which was generated from the treatment of hydrogel X with $E_2$ at a concentration of 250 μM, still exhibits the characteristic mechanical properties of a hydrogel. However, increasing the concentration of $E_2$, leads to softer hydrogels, and at a concentration of 500 μM, the system reaches a solution state (G' = 12.5 Pa; G" = 2.6 Pa), as shown in Supplementary Fig. 3. The time-dependent mechanical property changes of the $E_1$-stimulated or $E_2$-stimulated transition of hydrogel X to hydrogel Y or Z, respectively, are presented in Supplementary Fig. 4. The results indicate that G' values increase or decrease in the presence of $E_1$ or $E_2$ and reach saturation values after ca. 2 h. Furthermore, the $E_1$/$E_1$'-stimulated transitions of hydrogel X to Y and back are further supported by following shape transitions, which also proceed on a timescale of 2 h, *vide*

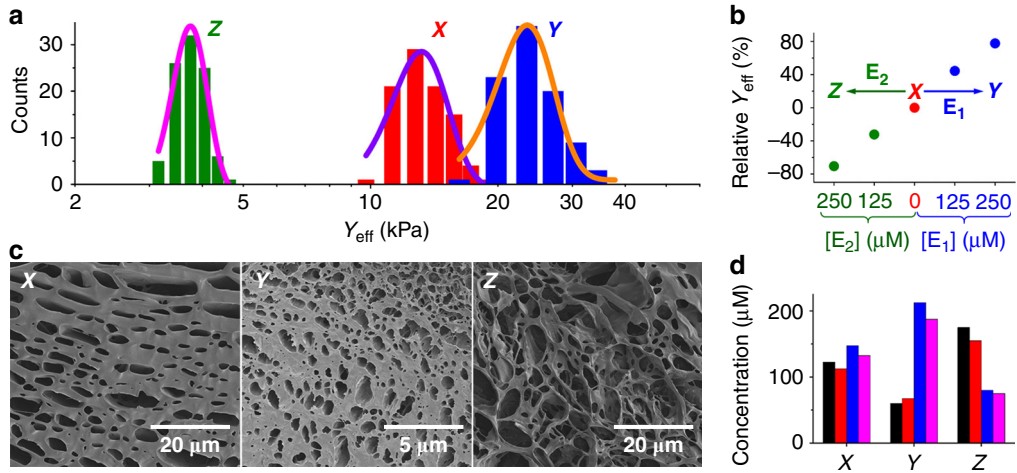

**Fig. 2** Controlling the stiffness properties of hydrogels consisting of triggered constitutional dynamic networks. **a** Histograms and corresponding Gaussian fits of Young's moduli associated with CDN hydrogels X, Y, and Z. **b** Young's modulus changes of hydrogel X programmed by the concentrations of effector $E_1$ or $E_2$. Relative $Y_{eff} = (Y_{eff} - Y_{eff(X)})/Y_{eff(X)} \times 100\%$, where $Y_{eff(X)}$ represents Young's modulus of hydrogel X without any triggers. **c** Scanning electron microscopy images of CDN hydrogels X, Y, and Z. **d** The concentrations of the constituents AA', BB', AB', and BA' in CDNs X, Y, and Z

| Table 1 Quantitative assessment of the compositions of the CDNs | | | | | |
|---|---|---|---|---|---|
| **System** | **Concentration (μM)** | | | | **Ratio of bidentate/ tetradentate[a]** |
| | **[AA']** | **[BB']** | **[AB']** | **[BA']** | |
| i | 122 | 112 | 147 | 132 | 1.68 |
| ii | 60 | 67 | 212 | 187 | 0.64 |
| iii | 175 | 155 | 80 | 75 | 4.26 |
| iv | 104 | 130 | 156 | 139 | 1.59 |
| v | 97 | 128 | 147 | 155 | 1.49 |
| Concentrations of the constituents and the ratios of the bidentate units (AA' and BB') and the tetradentate unit (AB'/BA') in the CDNs: (i) CDN X; (ii) CDN Y; (iii) CDN Z; (iv) The regenerated CDN X from CDN Y; (v) The regenerated CDN X from CDN Z | | | | | |
| [a]The ratio of bidentate/tetradentate was calculated by ([AA'] + [BB'])/{([AB'] + [BA'])/2} | | | | | |

*infra*. Scanning electron microscopy images of the different hydrogels (Fig. 2c) further support the relation between the degree of crosslinking of the CDN hydrogels and their stiffness properties. Hydrogel X exhibiting medium stiffness shows a medium density of large pores, hydrogel Y demonstrating the highest stiffness shows a higher density of small pores, while hydrogel Z revealing the lowest stiffness shows very large pores consistent with a low degree of crosslinking (*cf.* Fig. 1). A further step in the characterization of CDN hydrogels X, Y, and Z, has involved the quantitative assessment of the contents of the constituents in the CDNs. This was accomplished by examining the fluorescence properties of two sets of fluorophore-modified model CDNs in solution that included the constituents AA', BB' AB', and BA'. In one set, AA' was labeled with the fluorophore Cy3, BB' was labeled with the fluorophore Cy5, and AB' was labeled with a FRET pair Cy3/Cy5. The second set included the labeling of AA' and BA' with the fluorophore FAM. By probing the fluorescence intensities of the different fluorophores in the two sets of the model CDNs, the quantitative evaluation of the contents of the constituents in CDNs X, Y, and Z is possible. (For the detailed photophysical, quantitative evaluation of the contents of the constituents in the CDNs, see Supplementary Figs. 5–12 and accompanying discussion). Figure 2d and Table 1 show the concentrations of the constituents in CDNs X, Y, and Z. In CDN X, the contents of all four constituents are comparable. Subjecting CDN X to $E_1$ leads to CDN Y, where the

concentrations of BA' and AB' are upregulated and the concentrations of AA' and BB' are downregulated. The subsequent treatment of the resulting CDN Y with the counter trigger $E_1$' restores the original CDN X (Supplementary Fig. 13). In addition, the $E_2$-triggered transition of CDN X to CDN Z involves the upregulation of AA' and BB' and the downregulation of AB' and BA'. Subjecting the resulting CDN Z to the counter trigger $E_2$' recovers the original CDN X (Supplementary Fig. 14). (For further support that the contents of the constituents in the model CDNs represent the compositions of the CDN hydrogels, see Supplementary Figs. 15 and 16 and accompany discussion). The evaluation of the compositions of CDNs X, Y, and Z is fully consistent with the stiffness properties of the toehold-guided formation of the hydrogels shown in Fig. 1, which are controlled by the relative contents of the crosslinking unit AB'/BA' in the different CDNs.

**Trigger-induced shape transitions of the CDN hydrogels**. The triggered control over the stiffness properties of the CDN hydrogels is, also, imaged visually. Figure 3a shows that the treatment of a triangle-shaped hydrogel X with $E_1$ results in a smaller triangle (hydrogel Y), consistent with the enhanced crosslinking. The further treatment of the resulting hydrogel Y with the counter trigger $E_1$' restores the larger triangle-shaped structure demonstrating the recovery of hydrogel X. The shape transitions of hydrogels (X → Y → X) reach constant dimensions after being subjected to $E_1$ or $E_1$' for 2 h, respectively. Figure 3b shows the effect of trigger $E_2$ on the triangle-shaped hydrogel X. A shapeless quasi-liquid hydrogel is obtained, consistent with the formation of the low-stiffness hydrogel Z. The shapeless quasi-liquid hydrogel Z does not recover to the triangle shape upon the application of the counter trigger $E_2$'. Nonetheless, the resulting hydrogel is in a shrunken state consistent with the formation of a stiffer hydrogel X, Supplementary Fig. 17. In addition, subjecting the high-stiffness triangle-shaped hydrogel Y to the effectors $E_1$' and $E_2$ leads to its transition to the shapeless, quasi-liquid state Z, Supplementary Fig. 18. It should be noted that the shape transitions originate from swelling (hydration) effect dictated by the crosslinking of the respective hydrogel.

**CDN hydrogels for self-healing and bi-enzyme cascades**. The control over the stiffness properties of the CDN hydrogels paves

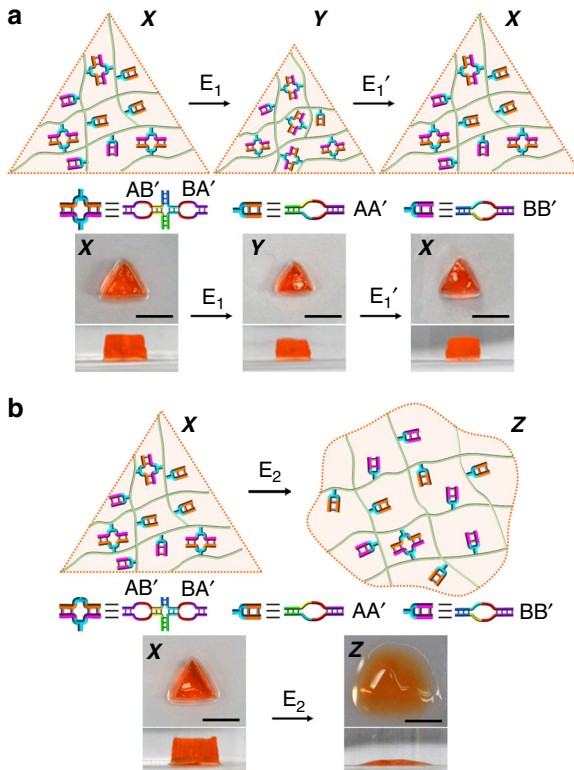

**Fig. 3** Visual imaging of the stiffness changes of the triggered CDN hydrogels. **a** The $E_1$-induced transition of the medium-stiffness CDN hydrogel X to the stiffer CDN hydrogel Y and the $E_1'$-stimulated reverse process. The transition of hydrogel X to the stiffer hydrogel Y is evident by the shrinkage of the triangle-shaped hydrogel. The $E_1'$-induced transition of hydrogel Y to X is evident by the regeneration of the original larger size of the triangle. **b** The $E_2$-induced transition of the triangle-shaped, medium-stiffness hydrogel X into the low-stiffness CDN hydrogel Z. Scale bars in all images correspond to 0.5 cm

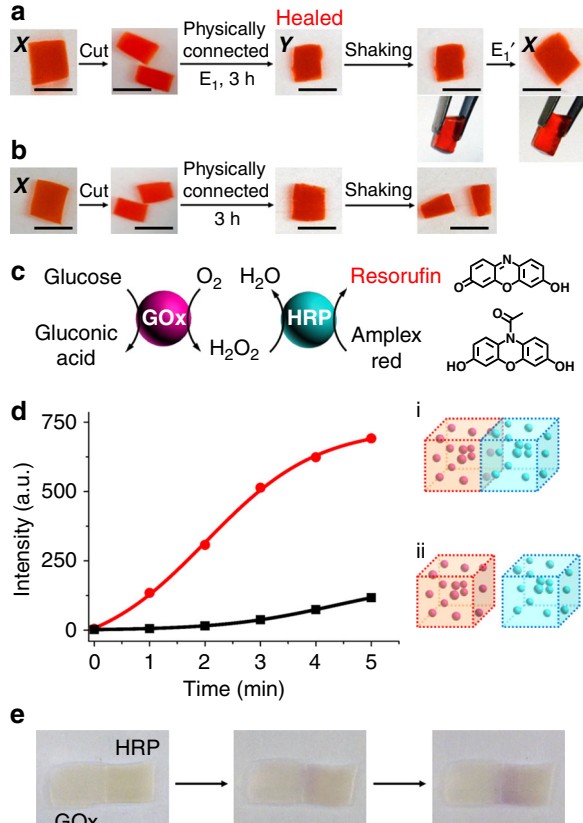

**Fig. 4** Self-healing of the CDN hydrogel matrices for a biocatalytic cascade. **a** Self-healing of CDN hydrogel X. The hydrogel was cut into two pieces which were physically interconnected. The $E_1$-induced transition of hydrogel X to the stiffer CDN hydrogel Y, yielded the healed hydrogel Y. Treatment of the healed hydrogel Y with $E_1'$ regenerated the healed hydrogel X. **b** A control experiment in the absence of trigger $E_1$ reveals that the $E_1$-induced transition of hydrogel X to Y is mandatory to yield the healed hydrogel matrix. **c** Operation of a bi-enzyme cascade consisting of glucose oxidase (GOx) coupled to horseradish peroxidase (HRP). The aerobic oxidation of glucose yields gluconic acid and $H_2O_2$. The generated $H_2O_2$ acts as a substrate for HRP that catalyzes the oxidation of Amplex Red to Resorufin. The latter product provides the fluorescence readout signal for the biocatalytic cascade. **d** Time-dependent fluorescence changes generated upon: (i) The operation of the GOx/HRP bi-enzyme cascade in a healed matrix composed of a GOx-loaded hydrogel matrix linked to a HRP-loaded hydrogel matrix; (ii) The operation of the bi-enzyme cascade in separated GOx-loaded hydrogel and HRP-loaded hydrogel matrices. **e** The color intensity changes of Resorufin generated by the bi-enzyme cascade reaction at the boundary of the healed hydrogel matrix. Scale bars in all images correspond to 0.5 cm

the way for emerging functions of the system. The first application of the CDN hydrogels included their use as self-healing materials, Fig. 4. Self-healing hydrogels attract growing interests as bio-adhesion and tissue engineering materials[36,37]. The self-healing of hydrogels using different auxiliary triggers, such as light[38–40], pH[41,42], redox reagents[43,44], or heat[45], was reported. To demonstrate the self-healing properties of the CDN hydrogels, a cuboid-shaped hydrogel X was cut into two pieces that were physically connected and subjected to trigger $E_1$ for 3 h (Fig. 4a). This process led to a healed shrunken hydrogel Y. The mended hydrogel Y behaved as an intact, healed hydrogel that could not be separated upon shaking (Supplementary Movie 1). It was switched, in the presence of the counter trigger $E_1'$, to the less-stiff, expanded hydrogel X that continued to behave as an intact healed matrix. In a control experiment, Fig. 4b, two physically connected pieces of hydrogel X were allowed to interact without $E_1$. After 3 h, only gentle shaking of the interconnected pieces resulted in their separation (Supplementary Movie 2), indicating that the healing process occurred only upon the stiffening of the interconnected pieces by transforming hydrogel X to hydrogel Y. The self-healing performance was further characterized by rheometric measurements, particularly by the induction of macroscopic cuts applying a high mechanical strain[46]. In this experiment, hydrogel X was initially subjected to a strain of 1%, leading to G' = 152.3 Pa and G" = 15.1 Pa. Subsequently, a strain corresponding to 200% was applied on the hydrogel to disrupt it. The resulting hydrogel was interacted with $E_1$ for 2 h to heal.

Afterwards, the healed matrix was treated with $E_1'$ to recover state X. The resulting healed matrix revealed G' = 125.9 Pa and G" = 18.7 Pa under a strain of 1%, indicating a ca. 83% recovery of the original mechanical strength of the hydrogel, Supplementary Fig. 19. Furthermore, it should be noted that the medium-stiffness hydrogel X and the low-stiffness hydrogel Z still include the functionalities to stimulate, in the presence of an appropriate mixture of effectors, their self-healing. Indeed, the physically connected pieces of hydrogels X and Z treated with $E_2'$ and $E_1$ for 4 h led to a healed hydrogel in state Y, Supplementary Fig. 20.

The self-healing property of the CDN hydrogel was, then, applied to construct an organized hydrogel assembly that activates a bi-enzyme cascade, Fig. 4c, d. The activation of biocatalytic cascades by the spatially proximate organization of

enzymes on macromolecular scaffolds[47,48] or in confined microenvironments[49] was reported. In addition, the programmed triggered release of enzymes from hydrogels led to the intercommunication of enzymes and the dictated activation of conjugated biocatalytic circuits[50]. Two CDN hydrogel X matrices (0.05 cm³) were prepared, where one was loaded with glucose oxidase, GOx, (~2.4 U, for the evaluation procedure, see Supplementary Fig. 21) and the other was loaded with horseradish peroxidase, HRP, (~2.9 U, for the evaluation procedure, see Supplementary Fig. 22). The two separated matrices underwent the self-healing process in the presence of $E_1$, and subsequently, the healed bi-enzyme hydrogel assembly in state Y was allowed to recover to state X in the presence of the counter trigger $E_1$'. The resulting hydrogel assembly was treated with glucose, in the presence of Amplex Red, to examine the activation of the enzymatic cascade shown in Fig. 4c. In this system, the aerobic GOx-catalyzed oxidation of glucose yields gluconic acid and $H_2O_2$. The diffusion of $H_2O_2$ into the HRP-containing hydrogel reservoir is then expected to stimulate the HRP-catalyzed oxidation of Amplex Red by $H_2O_2$ to form the fluorescent Resorufin. Figure 4d, curve i, and Supplementary Fig. 23a show the time-dependent fluorescence changes upon the operation of the bi-enzyme cascade in the healed, bi-enzyme hydrogel matrix. A control experiment (Supplementary Fig. 24) confirms that the leakage of GOx and HRP from the healed hydrogel matrix is negligible. In addition, the time-dependent fluorescence changes observed upon the activation of the bi-enzyme cascade in the presence of two separated hydrogel matrices (identical contents of GOx and HRP and volume of surrounding buffer solution), is displayed in Fig. 4d, curve ii, and Supplementary Fig. 23b. The healed bi-enzyme hydrogel matrix reveals a seven-fold enhanced activity of the bi-enzyme cascade as compared to that of the two separated matrices. That is, the efficient delivery of the GOx-generated $H_2O_2$ into the interconnected HRP-containing hydrogel reservoir leads to the activation of the enzymatic cascade. The operation of the bi-enzyme cascade in the healed two-hydrogel matrix can be visualized by following the color changes of the bi-enzyme generated Resorufin at the boundary of the healed hydrogels, Fig. 4e. The color of Resorufin in the HRP-loaded hydrogel intensifies with time and shows that $H_2O_2$ formed by GOx generates a concentration gradient upon diffusing into the HRP-loaded matrix. It should be noted that an efficient bi-enzyme cascade could be observed by the co-immobilization of GOx and HRP in one hydrogel matrix. Nonetheless, the results demonstrate that a self-healing process of two hydrogels loaded with two different enzymes leads to an effective biocatalytic cascade. In fact, the biocatalytic cascade in the healed matrix reveals an almost similar activity as the cascade proceeding in a single hydrogel matrix that contains the two enzymes, Supplementary Fig. 25. Such control over biocatalytic cascades by a self-healing process is important for in situ biomedical applications.

**Triggered release of loads from the CDN hydrogels**. In addition, stimuli-responsive hydrogel microcapsules acted as carriers for controlled drug release[51]. Accordingly, the control over the stiffness properties of the hydrogels by CDN composites was used to design CDN-controlled release matrices, Fig. 5a. CDN hydrogel X was loaded with mercaptopropionic acid-modified CdSe/ZnS quantum dots (QDs), and no release of the QDs could be detected. Subjecting the QDs-loaded hydrogel X to trigger $E_2$ resulted in the transition of hydrogel X to the quasi-liquid hydrogel Z. This process triggered-on the release of the QDs from the hydrogel, Fig. 5b. In another system, the anticancer drug doxorubicin was loaded in CDN hydrogel X. Inefficient release of the drug was observed without $E_2$, Fig. 5c. In fact, this release process occurred within 1 h, and afterwards, the release of the drug was blocked. Presumably, this inefficient release occurred from large pore domains that enabled the leakage of the entrapped drug. Treatment of the doxorubicin-loaded hydrogel X with $E_2$ resulted in the transition to the quasi-liquid hydrogel Z, and this stimulated the release of doxorubicin. Several control experiments were conducted to show the efficiency of the release

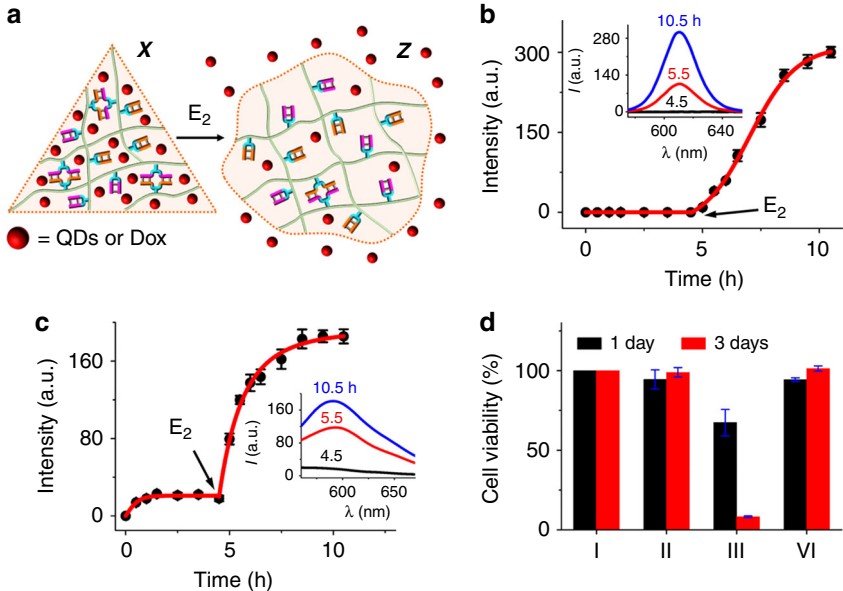

**Fig. 5** Triggered release of loads from the CDN hydrogels. **a** Schematic presentation of the release of CdSe/ZnS semiconductor quantum dots (QDs) or of doxorubicin (Dox) from CDN hydrogel X upon the $E_2$-stimulated transition of the medium-stiffness hydrogel X to the low-stiffness CDN hydrogel Z. **b** Time-dependent release profile of CdSe/ZnS QDs. **c** Time-dependent release profile of Dox. All data are presented as the mean ($n = 3$) and error bars in **b** and **c** represent standard deviation. **d** Cytotoxicity of the released anticancer drug Dox from hydrogel X without or with trigger $E_2$ toward MDA-MB-231 breast cancer cells. The cell viabilities (% of the control, entry I) are presented in the form of bars: (I) Untreated cells; (II) Cells treated with the released Dox from hydrogel X without $E_2$; (III) Cells treated with the released Dox from hydrogel X in the presence of $E_2$; (IV) Cells treated with the HEPES buffer

of doxorubicin from the hydrogels. For example, the release of doxorubicin from the high-stiffness hydrogel Y was examined, Supplementary Fig. 26. The release from hydrogel Y in the absence of triggers is substantially lower (ca. 3-fold lower) than that from hydrogel X, consistent with the higher stiffness and smaller pore-size of hydrogel Y. In addition, treatment of hydrogel Y with $E_1$' and $E_2$ leads to the transition of hydrogel Y to Z and to the effective release of the drug albeit the release rate is substantially slower compared to the $E_2$-triggered transition of hydrogel X to Z. The slower transition of hydrogel Y to Z, in the presence of triggers $E_1$' and $E_2$, is attributed to constrains associated with the dynamic separation of CDN Y by $E_1$' and the concomitant $E_2$-guided reassembly and reequilibration of the system to form hydrogel Z. We note, however, that while the release of the drug from hydrogel Z generated by the $E_2$-triggered transition of hydrogel X reached saturation after 5 h (Fig. 5c), the slower release of the drug upon the ($E_1$' + $E_2$)-triggered transition of hydrogel Y to Z did not lead to a saturated state even after 14 h (ca. 20% lower than the saturated fluorescence of the released drug, cf. Fig. 5c). In fact, after a substantially longer time (ca. 28 h), the release of the drug reached the saturation level. For further experiments characterizing the loading of doxorubicin and its release profile see Supplementary Figs. 27 and 28. The inefficient release of doxorubicin from hydrogel X and the $E_2$-triggered release of doxorubicin from hydrogel Z were then applied to examine the cytotoxicity of the released drug from the hydrogels towards MDA-MB-231 breast cancer cells, Fig. 5d. In these experiments, the doxorubicin-loaded hydrogel X was allowed to release doxorubicin in the absence of trigger $E_2$, or was subjected to trigger $E_2$ that resulted in the transition of hydrogel X to Z and the effective release of doxorubicin. The cytotoxic effects of the released doxorubicin on MDA-MB-231 cells were monitored after incubation of 1 and 3 days. The release from hydrogel X did not show significant cell death after 1 day or 3 days (II), while the release from hydrogel Z revealed impressive cytotoxicity toward MDA-MB-231 cells (III). After 1 day, ca. 30% cell death was observed, and 90% cell death was realized after 3 days. These results are consistent with the effective release of doxorubicin from hydrogel Z.

## Discussion

The present study introduces DNA-based adaptive constitutional dynamic networks as functional systems for the assembly of hydrogels exhibiting signal-triggered stiffness properties. The study demonstrates the adaptive, CDN-driven, reversibly switchable transitions of the hydrogels across three stiffness states that revealed intermediate-stiffness (CDN hydrogel X), high-stiffness (CDN hydrogel Y) and quasi-liquid, low-stiffness (CDN hydrogel Z) properties. Beyond the fundamental characterizations of the CDN hydrogels by microindentation, rheometry, SEM imaging and spectroscopic quantitative assessment of the contents of the constituents in the different CDN hydrogels, we discussed emerging material functions of the triggered CDN hydrogel systems. Specifically, we highlighted the dynamic self-healing properties of the hydrogels, the feasibility to apply self-healing functions to switch on biocatalytic cascades, and the possibility to use the triggered adaptive hydrogels for controlled release of loads.

## Methods

**Materials**. 4-(2-Hydroxyethyl)piperazine-1-ethanesulfonic acid sodium salt (HEPES), sodium chloride, magnesium chloride, glucose, Amplex Red, glucose oxidase (GOx, 124 U mg$^{-1}$), peroxidase from horseradish (HRP, average value 100 U mg$^{-1}$) and doxorubicin (Dox) were purchased from Sigma-Aldrich. "GelRed nucleic acid gel stain" was purchased from Invitrogen. CdSe/ZnS quantum dots in toluene were purchased from Evident Technologies and modified with 3-

Mercaptopropionic acid to yield mercaptopropionic acid-modified CdSe/ZnS quantum dots (QDs). The HEPES buffer used in all the experiments includes 10 mM HEPES, 50 mM NaCl and 20 mM MgCl$_2$ with a pH value of 7.2. Ultrapure water from NANOpure Diamond (Barnstead) source was used in all the experiments. The oligonucleic acid sequences used in the study include the following:

(**1**) A: 5'-CACGCGTCCAACCAGCCGTCGAAGCACCCAAAAAAACCAC AGTCCAGCAC-3'
(**2**) A': 5'-TGTGCTGGACTCCACACGAAAACTCACCTTCGACGGCTGG ACTGACAGAGGCAGAGAGCGCAC-3'
(**3**) B: 5'-CAGCCACGACTGCGTCAGTTCAGCGTGAGTTAAGATACAT TCACCAGCAC-3'
(**4**) B': 5'-GTGCTGGTGAGAGAGAGAATTGGGTGGCTGAACTGACGCTC CTGCTCCACGTGCGCTCTCTG-3'
(**5**) $L_1$: 5'-CCTCTGTCAGTCGTTGGACGCGTGGACTCGTTGGTG-3'
(**6**) $L_2$: 5'-GTGGAGCAGGAGCAGTCGTGGCTGCACCAACGAGTC-3'
(**7**) $L_{1m}$: 5'-GTGCGCTCTCTGCCTCTGTCAGTCGTTGGACGCGTG-3'
(**8**) $L_{2m}$: 5'-CAGAGAGCGCACGTGGAGCAGGAGCAGTCGTGGCTG-3'
(**9**) $E_1$: 5'-GAATGTATCCCGTGTGGAG-3'
(**10**) $E_1$': 5'-CTCCACACGGGATACATTC-3'
(**11**) $E_2$: 5'-ACCCAATTCACTTAACTCA-3'
(**12**) $E_2$': 5'-TGAGTTAAGTGAATTGGGT-3'
(**13**) A-Cy3: 5'-CACGCGTCCAACCAGCCGTCGAAGCACCCAAAAAAACC ACAGTCCAGCAC-Cy3-3'
(**14**) A'-FAM: 5'-FAM-TGTGCTGGACTCCACACGAAAACTCACCTTCG ACGGCTGGACTGA CAGAGGCAGAGAGCGCAC-3'
(**15**) B'-Cy5: 5'-Cy5-GTGCTGGTGAGAGAGAGAATTGGGTGGCTGAACT GACGCTCCTGCT CCACGTGCGCTCTCTG-3'

**Instruments and measurements**. Fluorescence spectra were recorded with a Cary Eclipse Fluorometer (Varian Inc.). The excitations of FAM, Cy3, Cy5, Amplex Red, CdSe/ZnS, and doxorubicin were performed at 450 nm, 545 nm, 620 nm, 570 nm, 588 nm, and 480 nm, respectively. The emissions of FAM, Cy3, Cy5, Resorufin, CdSe/ZnS and doxorubicin were recorded at 520 nm, 565 nm, 665 nm, 588 nm, 610 nm, and 590 nm, respectively. The absorbance spectra were recorded by a UV-2450 spectrophotometer (Shimadzu). Scanning electron microscopy (SEM) images were taken by using Extra High-Resolution Scanning Electron Microscope Magellan (TM) 400 L, microscope. The hydrogel samples were lyophilized previous to SEM imaging. Young's moduli of hydrogels were measured with a Piuma Nanoindenter (Optics 11, Amsterdam, NL). Mechanical properties were measured by a HAAKE MARS III rheometer (Thermo Scientific).

**Preparation of CDN hydrogel X**. A mixture of A, A', B, B', $L_1$ and $L_2$, 250 μM each, in HEPES buffer, was annealed at 90 °C for 5 min and then cooled down to 25 °C in a water bath, to yield CDN hydrogel X.

**Reversible transition of CDN hydrogel X to Y and back**. CDN hydrogel X was subjected to trigger $E_1$ (250 μM). The mixture was annealed at 90 °C for 5 min and then cooled down to 25 °C in a water bath, to yield CDN hydrogel Y. For the reverse transition to hydrogel X, the resulting hydrogel Y was treated with the counter trigger $E_1$' (250 μM). The mixture was annealed at 90 °C for 5 min and then cooled down to 25 °C in a water bath, to restore hydrogel X.

**Reversible transition of CDN hydrogel X to Z and back**. CDN hydrogel X was subjected to trigger $E_2$ (250 μM). The mixture was annealed at 90 °C for 5 min and then cooled down to 25 °C in a water bath, to yield CDN hydrogel Z. For the reverse transition to hydrogel X, the resulting hydrogel Z was treated with the counter trigger $E_2$' (250 μM). The mixture was annealed at 90 °C for 5 min and then cooled down to 25 °C in a water bath, to recover hydrogel X.

**Stiffness (Young's moduli) of the CDN hydrogels**. Young's moduli of hydrogels were measured with a Piuma Nanoindenter (Optics 11, Amsterdam, NL) and a probe with a spherical indenter tip with a radius of 9 μm and a stiffness of 0.45 N/m. Young's modulus was derived using the Oliver and Pharr fitting method,[52] with a $P_{min}$ of 65%, a $P_{max}$ of 85%. All Young's moduli given in the text refer to the effective Young's moduli ($Y_{eff}$). Values are given as the mean of about 100 indentations and the standard error of the mean (SEM).

**Bulk mechanical properties of the CDN hydrogels**. Mechanical properties were analyzed by a HAAKE MARS III rheometer (Thermo Scientific) with a parallel plate measuring geometry. Hydrogel samples (350 μL) were measured at a constant temperature of 20 °C, a strain of 1%, and a frequency of 1 Hz, if not stated otherwise. The gap was set to 1.8 mm.

**Compositions of CDN hydrogels X, Y, and Z**. Model CDN $X_m$ is taken as an example. In one set of experiment, a mixture of A-Cy3, A', B, B'-Cy5, $L_{1m}$ and $L_{2m}$, 250 μM each, in HEPES buffer, was annealed at 90 °C for 5 min and then cooled down to 25 °C in a water bath, to yield the Cy3/Cy5-labeled CDN $X_m$. One

microliter of the mixture was withdrawn and treated with 249 μL HEPES buffer. Subsequently, the fluorescence spectra of the Cy3/Cy5-labeled CDN $X_m$ were followed at $\lambda_{ex} = 545$ nm and $\lambda_{ex} = 620$ nm. In the other set of experiment, the FAM-labeled CDN $X_m$ was prepared following the above procedures, where A, A'-FAM and B' were used instead of A-Cy3, A', and B'-Cy5, respectively. One microliter of the mixture was withdrawn and treated with 249 μL HEPES buffer. Subsequently, the fluorescence spectrum of the FAM-labeled CDN $X_m$ was followed at $\lambda_{ex} = 450$ nm. Using the appropriate calibration curves corresponding to the fluorescence spectra of the individual fluorophore-labeled constituents (Supplementary Figs. 7, 8, and 11), the concentrations of the constituents in CDN $X_m$ were evaluated.

For the fluorophore-labeled CDN hydrogel Z: In one set of experiment, a mixture of A-Cy3, A', B, B'-Cy5, $L_1$ and $L_2$, 250 μM each, in HEPES buffer, was annealed at 90 °C for 5 min and then cooled down to 25 °C in a water bath, to yield the Cy3/Cy5-labeled CDN hydrogel X. The obtained hydrogel X was subjected to trigger $E_2$ (250 μM), annealed at 90 °C for 5 min and then cooled down to 25 °C in a water bath, to yield the Cy3/Cy5-labeled CDN hydrogel Z. After a 1: 250 dilution with HEPES buffer under shaking, the fluorescence spectra of the Cy3/Cy5-labeled CDN hydrogel Z were followed at $\lambda_{ex} = 545$ nm and $\lambda_{ex} = 620$ nm. In the other set of experiment, the FAM-labeled CDN hydrogel Z was prepared following the above procedures, where A, A'-FAM and B' were used instead of A-Cy3, A', and B'-Cy5, respectively. After a 1: 250 dilution with HEPES buffer under shaking, the fluorescence spectrum of the FAM-labeled CDN hydrogel Z was followed at $\lambda_{ex} = 450$ nm. Using the appropriate calibration curves corresponding to the fluorescence spectra of the individual fluorophore-labeled constituents (Supplementary Figs. 7, 8, and 11), the concentrations of the constituents in CDN hydrogel Z were evaluated.

**Trigger-induced shape transitions of the CDN hydrogels**. A mixture (100 μL) of A, A', B, B', $L_1$, and $L_2$, 250 μM each, in HEPES buffer, was annealed at 90 °C for 5 min and then quickly transferred into a triangle-shaped polytetrafluoroethylene mold. After staying overnight, the triangle-shaped hydrogel was extruded from the mold. It should be noted that the hydrogel was stained with GelRed.

For the transitions of CDN hydrogel X to hydrogel Y and back, the triangle-shaped hydrogel X was treated with trigger $E_1$ (100 μL of 0.8 mM in HEPES buffer) for 2 h. Then the surrounding solution was removed, and the resulting hydrogel was washed four times with 500 μL HEPES buffer to yield a smaller-sized and stiffer triangle-shaped hydrogel Y. After the treatment of the resulting hydrogel Y with the counter trigger $E_1'$ (100 μL of 0.8 mM in HEPES buffer), the recovery of hydrogel X occurred within 2 h. After removing the surrounding solution and washing four times with 500 μL HEPES buffer, no noticeable changes of the regenerated hydrogel X were observed.

For the transition of CDN hydrogel X to hydrogel Z, the prepared triangle-shaped hydrogel X was treated with trigger $E_2$ (100 μL of 0.8 mM in HEPES buffer) for 2 h, leading to a low-stiffness, shapeless hydrogel Z. After removing the surrounding solution and washing four times with 500 μL HEPES buffer, no noticeable changes of the resulting hydrogel Z were observed.

**Trigger-induced self-healing of CDN hydrogel X**. A mixture (200 μL) of A, A', B, B', $L_1$, and $L_2$, 250 μM each, in HEPES buffer, was annealed at 90 °C for 5 min and then quickly transferred into a cube-shaped polytetrafluoroethylene mold. It should be noted that the hydrogel was stained with GelRed. After staying overnight, the cuboid-shaped hydrogel was extruded from the mold and cut into two pieces. Subsequently, the two pieces of the hydrogel were physically connected and subjected to trigger $E_1$ (100 μL of 1.2 mM in HEPES buffer) for 3 h. The surrounding solution was removed, and the resulting hydrogel was washed four times with 500 μL HEPES buffer. A healed intact hydrogel Y was obtained and could not be separated upon shaking. Then the healed hydrogel was subjected to the counter trigger $E_1'$ (100 μL of 1.2 mM in HEPES buffer) for 2 h to restore the intact medium-stiffness hydrogel X.

For comparison, two pieces of hydrogel X were prepared following the above procedure. Subsequently, the two pieces of the hydrogel were physically connected and subjected to 100 μL HEPES buffer. After 3 h, only gentle shaking of the two pieces resulted in their separation.

**Bi-enzyme cascade guided by the healed hydrogel matrix**. For the GOx-loaded hydrogel X matrix, a mixture (200 μL) of A, A', B, B', $L_1$, $L_2$, 250 μM each, and GOx, 0.375 mg, in HEPES buffer, was annealed at 60 °C for 5 min and then quickly transferred into a cube-shaped polytetrafluoroethylene mold. After staying overnight, the resulting hydrogel cuboid was extruded from the mold, washed four times with 500 μL HEPES buffer (for the determination of the content of GOx loading in the hydrogel matrix, see Supplementary Fig. 21) and cut into four equal pieces (one of the pieces was used for self-healing and the activation of bi-enzyme cascade reaction).

For the HRP-loaded hydrogel X matrix, a mixture (200 μL) of A, A', B, B', $L_1$, $L_2$, 250 μM each, and HRP, 0.375 mg, in HEPES buffer, was annealed at 60 °C for 5 min and then quickly transferred into a cube-shaped polytetrafluoroethylene mold. After staying overnight, the resulting hydrogel cuboid was extruded from the mold, washed four times with 500 μL HEPES buffer (for the determination of the content of HRP loading in the hydrogel matrix, see Supplementary Fig. 22) and cut into four equal pieces (one of the pieces was used for self-healing and the activation of bi-enzyme cascade reaction).

The as prepared GOx-loaded hydrogel X matrix and the prepared HRP-loaded hydrogel X matrix were physically connected and subjected to trigger $E_1$ (100 μL of 0.8 mM in HEPES buffer) for 3 h. The surrounding solution was removed, and the resulting hydrogel matrix was washed four times with 500 μL HEPES buffer. Then the healed hydrogel matrix was subjected to the counter trigger $E_1'$ (100 μL of 0.8 mM in HEPES buffer) to restore the healed medium-stiffness hydrogel X matrix. After removing the surrounding solution and washing four times with 500 μL HEPES buffer, a healed GOx/HRP-loaded hydrogel X matrix was obtained.

To determine the leakage of enzymes from the healed GOx/HRP-loaded hydrogel X matrix, the healed matrix was subjected to 2 mL HEPES buffer and incubated for 1 h under gentle shaking. The incubation buffer was collected and subjected to glucose (3 μL of 1 M) and Amplex Red (3 μL of 20 mM) under gentle shaking. At different time intervals, aliquots of 60 μL of the solution were withdrawn and their fluorescence spectra were followed (Supplementary Fig. 24) implying negligible leakage of GOx/HRP from the healed hydrogel matrix.

The healed GOx/HRP-loaded hydrogel X matrix was subjected to 2 mL HEPES buffer solution including glucose (3 μL of 1 M) and Amplex Red (3 μL of 20 mM) under gentle shaking. At different time intervals, aliquots of 60 μL of the solution were withdrawn and their fluorescence spectra were recorded (Supplementary Fig. 23a).

For the control experiment of the non-linked (separated) hydrogel matrices, the as prepared GOx-loaded hydrogel X matrix and HRP-loaded hydrogel X matrix were subjected to 2 mL glucose (3 μL of 1 M) and Amplex Red (3 μL of 20 mM) under gentle shaking. At different time intervals, aliquots of 60 μL of the solution were withdrawn and their fluorescence spectra were followed (Supplementary Fig. 23b).

For the visualization of the bi-enzyme cascaded reaction at the boundary of the healed matrix (Fig. 4e), the healed GOx/HRP-loaded hydrogel X matrix (0.1 cm³ for each hydrogel) was subjected to 2 mL HEPES buffer solution including glucose (3 μL of 0.2 M) and Amplex Red (3 μL of 4 mM).

For the preparation of a single hydrogel X matrix loaded with GOx and HRP, a mixture (200 μL) of A, A', B, B', $L_1$, $L_2$, 250 μM each, GOx, 0.19 mg, and HRP, 0.19 mg, in HEPES buffer, was annealed at 60 °C for 5 min and then quickly transferred into a cube-shaped polytetrafluoroethylene mold. After staying overnight, the resulting hydrogel cuboid was extruded from the mold, washed four times with 500 μL HEPES buffer and cut into two equal pieces (one of the pieces was used for the bi-enzyme cascade reaction).

**Triggered release of loads from the CDN hydrogels**. For the preparation of the CdSe/ZnS QDs-loaded or doxorubicin-loaded hydrogel matrices, a mixture (200 μL) of A, A', B, B', $L_1$, $L_2$, 250 μM each, and CdSe/ZnS QDs (6 mM, 1 μL) or doxorubicin (10 mM, 1 μL), in HEPES buffer, was annealed at 60 °C for 5 min and then quickly transferred into a cube-shaped polytetrafluoroethylene mold. After staying overnight, the extruded resulting hydrogel matrices were washed four times with 500 μL HEPES buffer.

For the release of CdSe/ZnS QDs or doxorubicin from the CdSe/ZnS QDs-loaded or doxorubicin-loaded hydrogel matrices, the as prepared hydrogel matrices were subjected to 2 mL HEPES buffer and incubated for 4 h under gentle shaking. Then the mixtures were treated with trigger $E_2$ (20 μL of 4 mM) to trigger-on the release process. At different time intervals, aliquots of 60 μL of the solution were withdrawn and their fluorescence spectra were followed.

**Cell culture and cell viability experiments**. Human breast cancer cells (MDA-MB-231) were grown in 5% $CO_2$ RPMI-1640 medium supplemented with 10% FCS, L-glutamine, and antibiotics (Biological Industries). Mycoplasma contamination was periodically checked using mycoplasma detection kit (Cat B39032) from Tivan Biotech LTD in USA and was negative. Authentication of cells was checked using Microscope for morphology check and growth curve analysis.

Cells were plated one day prior to the experiment on 96-well plates for cell viability. Cell viability was assayed after incubation with the released doxorubicin from hydrogel X without or with trigger $E_2$ in MDA-MB-231 cells planted at a density of $1.2 \times 10^4$ cells per well in 96-well plates. After incubation of 1 day or 3 days, cells were washed intensively and then the cell viability was determined with the fluorescent redox probe, Presto-Blue. The fluorescence of Presto-Blue was recorded on a plate-reader (Tecan Safire) after incubation of 1 h at 37 °C ($\lambda_{ex} = 560$ nm; $\lambda_{em} = 590$ nm).

**Reporting summary**. Further information on research design is available in the Nature Research Reporting Summary linked to this article.

## Data availability
The data that support the findings of this study are available from the corresponding author upon reasonable request.

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

## Acknowledgements

This research was supported by the Israel Science Foundation and by the Minerva Center for Biohybrid Complex Systems. We thank Dr. Yang Sung Sohn and Prof. Rachel Nechushtai for the cell experiments.

## Author contributions

L.Y. and S.W. planned the nucleic acid sequences and performed the experiments. V.W. performed the microindentation experiments. I.W. mentored the project. All authors participated in the formulation of the paper.

## Competing interests

The authors declare no competing interests.
