## [Peer Review File · Nature Communications]

Reviewers' comments:

Reviewer #1 (Remarks to the Author):

In this manuscript, Willner and co-workers report on the stimuli responsive DNA hydrogel by constitutional dynamic network (CDN). By reversibly increasing and decreasing the tetradentate crosslinking units in the DNA hydrogel via CDN, they can switch the stiffness of the hydrogel. Additionally, self-healing is shown. They further used this healing strategy to join two pieces of gels loaded with GOX and HRP together and showed a bienzyme cascade reaction. Overall, this stimuli responsive hydrogel by DNA based CDN is interesting. However, one should also think why this kind of hydrogel should be made. What are the significant advantages of this hydrogel compared to the many examples from the Willner group and other hydrogels showing stimuli responsiveness eg. triplex, G-quadruplexes, i-motif, and aptamers. While the DNA crosslink changes alter the stiffness of the gel, the diffusion of the effectors limits rapid switching and also a spatial control can hardly be imposed. Also such cascade reaction, self-healing and shape recovery have been shown repeatedly for similar gels. In summary, while the manuscript shows a new control mechanism, the overall novelty of this paper is not enough for Nature Communications. More detailed comments are listed below.

1. By adding the E1 or E2 guiding strands, they can deswell and swell the hydrogel. Then, how fast the hydrogel can respond to guiding strands. Are there any time dependent stiffness measurements after adding E1 and E2 as well as for the reverse process? This is of course size dependent, and measuring bulk moduli using rheology would be much more appropriate rather than only measuring surface values, where changes occur quickly.
2. The experimental protocol of the indentation is incomplete (machine, tip type etc)
3. After the addition of E2, does the hydrogel completely converted to the sol state or is it still a hydrogel. What are the storage modulus G' and loss modulus G'' .
4. Is it possible to get more programmable stiffness by varying the E1 or E2 amount?
5. Although the authors argued that the measured constituents of the crosslinking units in hydrogel Z is identical to model CDN hydrogel Z_m , one still cannot get any conclusion from these models for the constituents in hydrogel X and Y. Hydrogel Z might be almost in sol state; thus, it shows similar results to the model. However, hydrogel X and Y are still solid gel. The reaction should still be less efficient compared to that for the solution state. G' and G'' data of hydrogel Z might further prove this comment.
6. For the self-healing part, how much mechanical strength can be recovered after the healing? And how much time is needed to recover as much mechanical strength as possible?
7. The bienzyme cascade part: Is it possible to visualize the production generation at the boundary of the two pieces gels. It would be nice to show some color gradients, which will in turn prove the direct diffusion of H_2O_2 from GOX loaded gel to HRP loaded gel through the hydrogel phase. What is the diffusion difference through the hydrogel compared to that through aqueous solution?
8. The authors showed some controlled drug release experiment but this is quite an old model for controlled release via stimuli responsive hydrogel. There is also no in vitro/vivo drug delivery investigation.

Reviewer #2 (Remarks to the Author):

Summary

The authors report the production of reversible switch DNA hydrogel matrix depending on the stiffness rates. The stiffness was tuned by altering the tetradentate and bidentate constituents within the polymer network. The tetradentate gives a stronger binding in the network, therefore resulting in the enhanced mechanical strength and stiffness of the gel, while the bidentate has the opposite effect. The effector stabilizing the tetradentate constituents results in higher stiffness of the produced hydrogel due to the upregulated tetradentate moieties. Lower stiffness hydrogels are produced due to

the orthogonal effectors which stabilized the bidentate molecules. By fine tuning the concentration of the effectors which supports the tetradentate and bidentate molecules, hydrogels of various stiffness could be achieved with the property of reversibility. They also showed the self-healing property and release of small molecules such as the drug doxorubicin/QDs from the CDN X hydrogels through the trigger.

Minor corrections

1. Citations for Figure 5 and Supplementary Figure 13 is missing in the main text.
2. Scale on the Y-axis for the Supplementary Figure 1 b and d should be changed
3. The scale bar is missing for Figure 4 a and b

Major Comments

For the transition of the hydrogel crosslink network, would there be a limit in using the trigger E2? As E2 trigger up-regulates the AA' and BB' concentration of the cross-link network, an "overexposure" to E2 trigger may transform too much of the cross-link into AA' and BB', and render the hydrogel liquid and unable to remain the gel form. Thus, it would be interesting to learn how much E2 trigger is needed to dissociate the gels.

In Figure 3. The authors show the reversible property over the stiffness measures of the CDN hydrogels through visual characteristics. They showed that the treatment of the CDN hydrogel Y with the counter trigger E'1 restores the original shape of the CDN hydrogel X. Whether the CDN hydrogel Z restores its original shape (CDN hydrogel X) when using the trigger E'2 from CDN hydrogel Y? Gaining the shape of the hydrogel from this point through trigger E'2 would be more interesting and add significant meaning to the shape restoration properties of the quasi-liquid CDN hydrogel. This also gives more clue about the shape memory of the CDN hydrogels.

The transition of shape change contributing for the stiffness properties of these CDN hydrogels is undoubtedly interesting (Figure 3). However, the authors show the reversible property of the CDN hydrogels only from X to Y and then to X. Can the authors do an experiment where they could change the shape of the CDN hydrogel X to CDN hydrogel Y and then to CDN hydrogel Z (i.e., X Y Z)? For the reversibility of the stiffness switch, is there a limit to the switch? Say, will the gels dissociates or maintain stiffness after certain cycles? The stiffness may start to have some weird result if the triggers remain clung to the gels, and is more possible to occur as the cycle increases. Moreover, from studying the data of Supplementary Figure 1 d (and also d), the stiffness after one cycle shows a slight decrease. Thus, to better understand the reversibility of the process, data data of showing the switching stiffness more than one cycle is needed to understand if there would be a continuous decrease in stiffness for the gel.

For self-healing experiments, there is an interesting questions: are hydrogels of different CDN able to heal with each others? Hydrogels with different CDN still share the same crosslink sequences (though different structure), and treating them with same trigger should ultimately push the crosslinks to certain structures. As long as there is some structures changing around the interface between two gels, they should be able to form bounds on the interface and heal with each other. However, the bonds formed may or may not be sufficient to hold the two gels. Still, this would be a great experiments in testing the limit of the self-healing property of the hydrogel.

In figure 4a and b, the authors show the self-healing property of the CDN hydrogels through the DNA triggers. In the control they showed in the absence of the trigger the hydrogels could not able to join by themselves. The readers would better understand if they provide a video file for this particular experiment showing the self-healing property of the CDN hydrogel through the addition of the trigger. The authors show the self-healing mechanism of the hydrogels stimulates a biocatalytic mechanism thereby increasing the fluorescence of the produced dye Resorufin. In Figure 4a Yui et al, showed that the self-healing nature of the CDN hydrogel by having two hydrogels and then healing the two to one through the addition of trigger E1. They prepared 2 CDN X hydrogels one containing glucose oxidase and other containing horseradish peroxidase. The self-healing property of the hydrogels is triggered through E1 resulting in the newly assembled CDN Y hydrogel which is essential for the bio-catalytic cascade reaction. The control experiments were also appropriate determining the cascade reactions takes place only with the healed hydrogel.

Why did the authors use two separate CDN hydrogels for loading of the catalytic contents, namely glucose oxidase and horseradish peroxidase?

What would be the result if both the glucose oxidase and horseradish peroxidase are inoculated into one CDN Hydrogel X?

For controlled release experiments of releasing anticancer drug doxorubicin, inefficient release of the drug was observed when using hydrogels with CDN X (figure 5c). Hypothesis is that the inefficient release was caused by the large pore size of the gel. Thus, a simple experiment may be conducted to test the hypothesis, or, at least, the limit of the gel. As hydrogels with CDN Y have a higher degree of crosslinking, they should not only be stiffer than CDN X gels but also have smaller pore size. The authors may conduct experiments using CDN Y gels with doxorubicin entrapped to see if the pore size of the gels are smaller enough to contain the drug.

Experiments with doxorubicin and QDs need more attention. I feel this is too early to give a positive notion about the release of the drug molecules from the hydrogels.

In Page 8 (Probing the triggered...): The authors say that the hydrogels were treated with the trigger E1 for the release of doxorubicin/QDs, but in Figures 5b and c they display trigger E2 for the release of doxorubicin/QDs. This is confusing for the reader.

What type of release profile is exhibited for doxorubicin from the hydrogel? Is it controlled release? If so, can you explain this with a release-kinetics model such as first-order, zero-order, etc?

How does doxorubicin bind to the hydrogel, could the authors explain about the mechanism of the same? How do you confirm that the doxorubicin is in the DNA hydrogel? Is it possible for the authors to do SEM analysis showing the binding of the drug into the gel matrix?

The authors claim that the trigger E2 helps in the release profile of the drug doxorubicin within 1 hour. The large porous size of the hydrogel Z and also the shape change to quasi-liquid helps in the release of the drug. An effective control would be after loading the drug into CDN X hydrogels, instead of the trigger activation, could the authors cut the gel into two and check the release profiles?

Did the authors do the same experiment with the medium density pores, of CDN Y hydrogel? The authors claim that the CDN X hydrogel stimulated no release because of the small pore size. However, there should be a slow/steady increase in the release profile of the drug when the drug is loaded in CDN Y hydrogel.

Re: Ms. NCOMMS-19-13747-T

Title: "DNA-Based Constitutional Dynamic Network Hydrogels Exhibiting Switchable Stiffness, Self-Healing, Controlled Release and Matrix-Guided Biocatalytic Cascades"

Detailed specific response to the reviewers' comments

The following point-by-point listed corrections were introduced into the paper:

Reviewer #1:

We feel that the reviewer undervalued the significance and novelty of the present study. The reviewer is correct that many stimuli-responsive nucleic acid-based hydrogels of controlled stiffness were reported. Nonetheless, the present study introduces new concepts in designing stimuli-responsive DNA-based hydrogels. These include: (i) The demonstration that a constitutional dynamic network of nucleic acids can undergo triggered reconfiguration across three hydrogel stiffness states is unprecedented, and this concept may be extended to other materials and systems. (ii) The study has an important basic scientific significance as it provides an optical (fluorescence) means to correlate between the molecular compositions of the constituents in the different hydrogel states and the bulk stiffness properties of the respective hydrogels. (iii) The broad applications of DNA-based constitutional dynamic network hydrogels as self-healing materials, controlled release matrices, and their control over biocatalytic cascades are novel.

These unique contributions of the present study were further emphasized at the end of the "introduction" section.

Comment 1: *By adding the E1 or E2 guiding strands, they can deswell and swell the hydrogel. Then, how fast the hydrogel can respond to guiding strands. Are there any time dependent stiffness measurements after adding E1 and E2 as well as for the reverse process?.....*

Response: The reviewer's request to characterize the bulk mechanical properties of the different hydrogels by rheometry (beyond the microindentation results) was followed. These results were described in Supplementary Fig. 2, and discussed in the main text, p. 7.

In addition, the time-dependent transitions between hydrogels X to Y and X to Z were evaluated by rheometry. The results are presented in Supplementary Fig. 4, and discussed in the main text, p. 8.

Comment 2: *The experimental protocol of the indentation is incomplete (machine, tip type etc).*

Response: Further details on the indentation setup and experiments and a reference addressing the fitting procedure of the data were added to the experimental section, Supplementary Information.

Comment 3: *After the addition of E2, does the hydrogel completely converted to the sol state or is it still a hydrogel.....*

Response: Rheological experiments indicate that the hydrogel in state Z is still in a gel-state. The rheological results (G'/G'' values) are presented in Supplementary Fig. 2 and discussed in the main text, p. 7.

Comment 4: *Is it possible to get more programmable stiffness by varying the E1 or E2 amount?*

Response: Indeed, by controlling the concentrations of effectors E_1 and E_2 , programmability of the stiffness of the hydrogel can be achieved. The results are shown in Fig. 2b, and discussed in the paper, p. 7.

Comment 5:*The reaction should still be less efficient compared to that for the solution state. G' and G'' data of hydrogel Z might further prove this comment.*

Response: The G' and G'' values of hydrogel Z confirm that the material is, under the experimental conditions, in a soft hydrogel state. The values of G' and G'' are provided and the results were discussed in the main text, p. 7.

Comment 6: For the self-healing part, how much mechanical strength can be recovered after the healing? And how much time is needed to recover as much mechanical strength as possible?

Response: The experiments addressing the recovery of the mechanical strength of the hydrogel, after healing, were described in Supplementary Fig. 19. We found that after two hours of healing, ca. 83 % of the original mechanical strength of the hydrogel were recovered. The results of these experiments were discussed in the main text, p. 10.

Comment 7:*It would be nice to show some color gradients, which will in turn prove the direct diffusion of H_2O_2 from GOX loaded gel to HRP loaded gel through the hydrogel phase.....*

Response: Indeed, the operation of the bi-enzyme cascade can be followed by the visual detection of a color gradient at the boundary of the healed hydrogel pieces. These results are presented in Fig. 4e that clearly demonstrates the diffusion of H_2O_2 into the horseradish peroxidase-loaded hydrogel. The images were further discussed in the main text, p. 11.

Comment 8:*There is no in vitro/vivo drug delivery investigation.*

Response: We have examined the cytotoxicity of the doxorubicin released upon the E_2 -stimulated transition of hydrogel X to Z on MDA-MB-231 cancer cells, and compared the results to a set of control systems, Fig. 5d. Impressive 90% of MDA-MB-231 cell death was observed after an incubation time of 3 days, in which the cells were exposed to the doxorubicin released from hydrogel Z. The results were discussed in the main text, p. 13.

Reviewer #2:

Minor corrections

1. Citations for Figure 5 and Supplementary Figure 13 is missing in the main text.
2. Scale on the Y-axis for the Supplementary Figure 1 b and d should be changed.
3. The scale bar is missing for Figure 4 a and b.

Response: All minor corrections were addressed in the text.

Comment 1:It would be interesting to learn how much E_2 trigger is needed to dissociate the gels.

Response: The changes of the G'/G'' values as a function of the concentration of E_2 are presented in Supplementary Fig. 3. The concentration at which the hydrogel dissociates into a solution ($G' = 12.5$ Pa and $G'' = 2.6$ Pa), is ca. 500 μ M. This result was mentioned in the main text, p. 7.

Comment 2:*Gaining the shape of the hydrogel from this point through trigger E_2 would be more interesting and add significant meaning to the shape restoration properties of the quasi-liquid CDN hydrogel. This also gives more clue about the shape memory of the CDN hydrogels.*

Response: It should be noted that the presented hydrogels do not include any shape-memory code, so hydrogel Z cannot recover to the shaped triangle. The treatment of hydrogel Z with E_2

yields, as expected, a shrunken configuration, due to the formation of a less hydrated, higher-stiffness hydrogel in state X. These results are presented in Supplementary Fig. 17 and they were discussed in the main text, p. 9. It should be noted that all hydrogels in the present study **are not** shape-memory matrices, and all shape changes originate from stiffness/degree of hydration changes.

Comment 3:*Can the authors do an experiment where they could change the shape of the CDN hydrogel X to CDN hydrogel Y and then to CDN hydrogel Z (i.e., X Y Z)?*

Response: Indeed, the hydrogel transitions $X \rightarrow Y \rightarrow Z$ can be stimulated by the application of the appropriate effectors. This is demonstrated in Supplementary Fig. 18, and the results were discussed in the main text, p. 9. The reviewer should note, however, that the transition from hydrogel Y to Z is stimulated by two effectors E_1' and E_2 . Once hydrogel Z is formed, the stiffness/hydration features are fully reversible, but the triangle shape is lost, since the system lacks any shape-memory code.

Comment 4: *For the reversibility of the stiffness switch, is there a limit to the switch?..... data of showing the switching stiffness more than one cycle is needed.*

Response: The major cause for losing the reversible feature of the hydrogel is the continuous addition of the effector-solutions of E_1 and E_1' . In the original paper, the addition of the effectors E_1 and E_1' introduced a volume change (ca. 12%) of the hydrogel, and this resulted in the slight decrease in the reversibility of the stiffness property of the hydrogel. We now add experimental results, Supplementary Fig. 1e, showing that the introduction of small volumes of concentrated E_1 and E_1' (while retaining the molar contents of E_1 and E_1' in the hydrogel) allow the switching of the hydrogel across two cycles with only a ca. 3% decrease in the respective stiffness values. These results were discussed in the caption of Supplementary Fig. 1.

Comment 5: *are hydrogels of different CDN able to heal with each others? this would be a great experiments in testing the limit of the self-healing property of the hydrogel.*

Response: This is, indeed, an interesting question, and we thank the reviewer for the comment. We find that the interconnection of the two hydrogels consisting of CDN X and CDN Z can, indeed, self-heal due to the existence of complementary units of the boundary of the two hydrogels. The results are presented in Supplementary Fig. 20, and discussed in the main text, p. 10.

Comment 6: *The readers would better understand if they provide a video file for this particular experiment showing the self-healing property of the CDN hydrogel through the addition of the trigger.*

Response: As requested, two movie files (Supplementary Movies 1 and 2) that present the self-healing properties were added and mentioned in the main text, p. 10.

Comment 7: *Why did the authors use two separate CDN hydrogels for loading of the catalytics contents, namely glucose oxidase and horseradish peroxidase? What would be the result if both the glucose oxidase and horseradish peroxidase are inoculated into one CDN Hydrogel X?*

Response: The reviewer is correct that the integration of the two enzymes glucose oxidase and horseradish peroxidase in one hydrogel would allow the operation of the bi-enzyme cascades (such systems were previously reported by us and others). The purpose of the present study is, however, to demonstrate that self-healing of two pieces of hydrogels containing two different

enzymes still allows the operation of the bi-enzyme cascade. Such phenomenon could be important for future biomedical applications. We demonstrate that effective communication between the two pieces of enzyme-loaded self-healed hydrogel exists.

In fact, we find that the bi-enzyme cascade in the self-healed matrix reveals comparable activity to the system where the two enzymes exist in a single hydrogel matrix. These results are presented in the Supplementary Fig. 25, and discussed in the main text, p. 12.

Comment 8: *The authors may conduct experiments using CDN Y gels with doxorubicin entrapped to see if the pore size of the gels are smaller enough to contain the drug.*

Response: The requested experiment was performed, and the results are presented in Supplementary Fig. 26, and discussed in the main text, p. 12. As expected, the release of doxorubicin from hydrogel Y is very inefficient, consistent with the higher stiffness/smaller pore-size of the hydrogel.

Comment 9: *In Page 8 (Probing the triggered...): The authors say that the hydrogels were treated with the trigger E1 for the release of doxorubicin/QDs, but in Figures 5b and c they display trigger E2 for the release of doxorubicin/QDs.*

Response: The reviewer is correct that the presentation of the experimental section in the original Supplementary Information page 8 was confusing. This originated from a typo where instead of effector E₂, we mentioned effector E₁. This error is now corrected.

Comment 10: *What type of release profile is exhibited for doxorubicin from the hydrogel?*

Response: Although the formulation of a kinetic model for the release of the drug from E₂-stimulated transition of hydrogel X to Z is an interesting question, it is beyond the scope of the present study. The reviewer should realize that the complexity of the hydrogels, e.g. stiffness values, geometrical constraints and dimensions, diffusion constants and more, are anticipated to affect the kinetic release profile. These difficulties were added in Supplementary Information and mentioned in the text, p. 13.

Comment 11: *How do you confirm that the doxorubicin is in the DNA hydrogel? Is it possible for the authors to do SEM analysis showing the binding of the drug into the gel matrix?*

Response: Based on previous studies, the doxorubicin binds to DNA-based hydrogels through intercalation and/or H-bonding. SEM images suggested by the reviewer to confirm the binding of doxorubicin to the hydrogel are impossible due to the small size of the doxorubicin. We confirmed, however, the binding of doxorubicin to hydrogel X by confocal fluorescence microscopy imaging. The results are presented in Supplementary Fig. 27, and mentioned in the main text, p. 13. The results clearly show the characteristic green fluorescence of the doxorubicin in the hydrogel.

Comment 12: *An effective control would be after loading the drug into CDN X hydrogels, instead of the trigger activation, could the authors cut the gel into two and check the release profiles?*

Response: The comment of the review is interesting and the experiment suggested by the reviewer could shed light on the release mechanism. The results of the experiment suggested by the reviewer are presented in Supplementary Fig. 28 and accompanying discussion. We find that after ca. two hours, the release of the drug turns to be very slow. We attribute the fast release of doxorubicin within the first hour to the release from the thin hydrogel boundary associated with the edges of the hydrogels. Cutting the hydrogel into two equal pieces and examination of the

release of the drug from the cut hydrogel, Supplementary Fig. 28, reveals a triggered-on release of the drug. This is attributed to the formation of new release boundaries after the cutting process.

Comment 13: *However, there should be slow/steady increase in the release profile of the drug when the drug is loaded in CDN Y hydrogel.*

Response: The reviewer is correct that the reconfiguration of CDN hydrogel Y to Z in the presence of the effectors E₁' and E₂ should lead to a slower release of the doxorubicin drug compared to the E₂-triggered transition of hydrogel X to Z. The results are presented in Supplementary Fig. 26 and discussed in the text, p. 12.

REVIEWERS' COMMENTS:

Reviewer #1 (Remarks to the Author):

The authors made some relevant clarifications to the scope of the manuscript and the additional data is interesting and quantifies several previous claims. I can be convinced to get this MS accepted at Nature Communications.

Reviewer #2 (Remarks to the Author):

Review for DNA-Based Constitutional Dynamic Network Hydrogels Exhibiting Switchable Stiffness, Self-Healing, Controlled Release and Matrix-Guided Biocatalytic Cascades

The authors have satisfactorily answered the questions in the first version. I recommend this work for publication after the authors address the following questions and comments in the manuscript:

Comments and questions for the authors

For transition of the hydrogel from $X \rightarrow Y \rightarrow Z$, methods of transition from $Y \rightarrow Z$ should be different from $X \rightarrow Y$, as the latter takes a single inducer E1, while the former requires two, E1' and E2. Thus, method clarifying such treatment should be added whether in the method part of the main text or under the description of figure 18, specifying the concentration of each trigger strands and whether both strands are added to the system simultaneously or sequentially.

The self-healing ability of different hydrogels is very interesting! However, similarly, for the self-healing process, specific method should also be added, clarifying the concentration of the inducer strands and also whether the strands are added simultaneously or sequentially.

In the main text, p13, the authors mentioned that "The release from hydrogel Y in the absence of triggers is substantially lower than that from hydrogel X, consistent with the higher stiffness and smaller pore-size of hydrogel Y." Comparing the result of figure 5b and supplementary figure 26, for readers' better understanding, the authors should point out the quantity difference between two experiment to avoid confusion.

Also, comparing the result of figure 5b and supplementary figure 36, the readers can see that hydrogel Y has a lower efficiency for release as the final intensity of the released drugs is lower than that of hydrogel X. However, the two hydrogels, X and Y, should all transform into the same type of hydrogel (Z) in the end, and has the same final intensity.
Can the authors provide an explanation for this?

Are there difference between the two hydrogel Zs as they come from different hosts?

If so, where does the difference comes from?

For hydrogel Z from Y does it requires more E2 or both inducers to have properties similar to hydrogel Z from X ?

Re: NCOMMS-19-13747A

Title: “Stiffness-Switchable DNA-Based Constitutional Dynamic Network Hydrogels for Self-Healing and Matrix-Guided Controlled Chemical Processes”

Point-by-point responses to the reviewers' comments

The following changes/explanations were introduced into the paper and the point-by-point responses are as follows:

Reviewer #1:

The reviewer had no additional comments and recommended the acceptance of the paper as it is.

Reviewer #2:

Comment 1: *For transition of the hydrogel from $X \rightarrow Y \rightarrow Z$, methods of transition from $Y \rightarrow Z$ should be different from $X \rightarrow Y$, as the latter takes a single inducer E_1 , while the former requires two, E_1' and E_2 . Thus, method clarifying such treatment should be added whether in the method part of the main text or under the description of figure 18, specifying the concentration of each trigger strands and whether both strands are added to the system simultaneously or sequentially.*

Response: The concentrations of the effectors were provided in the caption of Supplementary Fig. 18. We further explained in the caption that the effectors E_1' and E_2 were added simultaneously to induce the transition of hydrogel Y to hydrogel Z.

Comment 2: *The self-healing ability of different hydrogels is very interesting! However, similarly, for the self-healing process, specific method should also be added, clarifying the concentration of the inducer strands and also whether the strands are added simultaneously or sequentially.*

Response: The concentrations of the effectors to induce the self-healing of hydrogels X and Z were introduced into the caption of Supplementary Fig. 20. In addition, we explained in the caption that the effectors E_1 and E_2' were added simultaneously to induce the self-healing process.

Comment 3: *In the main text, p13, the authors mentioned that “The release from hydrogel Y in the absence of triggers is substantially lower than that from hydrogel X, consistent with the higher stiffness and smaller pore-size of hydrogel Y.” Comparing the result of figure 5c and supplementary figure 26, for readers' better understanding, the authors should point out the quantity difference between two experiments to avoid confusion.*

Response: As requested, the difference in the release profiles shown in Fig. 5c and Supplementary Fig. 26 was quantified and the details were provided in the main text, p. 13.

Comment 4: *Also, comparing the result of figure 5c and supplementary figure 26, the readers can see that hydrogel Y has a lower efficiency for release as the final intensity of the released drugs is lower than that of hydrogel X. However, the two hydrogels, X and Y, should all transform into the same type of hydrogel (Z) in the end, and has the same final intensity. Can the authors provide an explanation for this?*

Are there difference between the two hydrogel Zs as they come from different hosts?

If so, where does the difference comes from?

For hydrogel Z from Y does it requires more E2 or both inducers to have properties similar to hydrogel Z from X ?

Response: Indeed, the transitions of hydrogel X to Z and of hydrogel Y to Z generate the same hydrogel Z product. The kinetic profiles for generating hydrogel Z from hydrogel X or Y are, however, different. An explanation for the origin of this difference was introduced into the main text, p. 13. The fluorescence intensity of the released drug upon the transition of hydrogel Y to Z, after a time interval of 14 hours, is indeed ca. 20% lower than that from the transition of hydrogel X to Z after a time interval of 5 hours. This 20% difference originates from incomplete transition of hydrogel Y to Z. After a time interval of 28 hours of the treatment of hydrogel Y with effectors E₁' and E₂, the fluorescence intensity of the released drug upon the transition of hydrogel Y to Z is similar to that from the E₂-guided transition of hydrogel X to Z. The results indicate that hydrogel Z generated by the two paths are identical. This information was added to the text, p. 13. In addition, the concentrations of effectors E₁' and E₂ required to induce the transition of hydrogel Y to Z were added to the caption of Supplementary Fig. 26.